

# Bistability and nonequilibrium condensation in a driven-dissipative Josephson array: A c-field model

Matthew T. Reeves[*] and Matthew J. Davis[†]

Australian Research Council Centre of Excellence in Future Low-Energy Electronics Technologies, School of Mathematics and Physics, University of Queensland, St Lucia, QLD 4072, Australia.

[*] m.reeves@uq.edu.au     [†] mdavis@physics.uq.edu.au

## Abstract

Developing theoretical models for nonequilibrium quantum systems poses significant challenges. Here we develop and study a multimode model of a driven-dissipative Josephson junction chain of atomic Bose-Einstein condensates, as realised in the experiment of Labouvie *et al.* [Phys. Rev. Lett. 116, 235302 (2016)]. The model is based on c-field theory, a beyond-mean-field approach to Bose-Einstein condensates that incorporates fluctuations due to finite temperature and dissipation. We find the c-field model is capable of capturing all key features of the nonequilibrium phase diagram, including bistability and a critical slowing down in the lower branch of the bistable region. Our model is closely related to the so-called Lugiato-Lefever equation, and thus establishes new connections between nonequilibrium dynamics of ultracold atoms with nonlinear optics, exciton-polariton superfluids, and driven damped sine-Gordon systems.

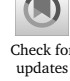

# 1   Introduction

Dissipation is often undesirable in the study of quantum systems, as it typically leads to the loss of quantum features such as coherence, entanglement and the washing out of interference. However, in recent years it has been realized that by adding controlled sources of dissipation to many-body quantum systems one can engineer novel quantum states of matter [1–6]. For example, an appropriate environment coupling can be harnessed to generate robust entangled states [7–9], or enable dissipative quantum computation protocols [10]. Further, the addition of driving allows the controlled study of quantum transport processes [11], $\mathcal{PT}$-symmetric quantum mechanics [12, 13], and dissipative phase transitions [14]. The ability to introduce engineered driving and dissipation offers the controlled study of nonequilibrium steady-states, which can exhibit emergent and exotic properties that cannot be achieved at or near equilibrium [15, 16]. Ultracold atomic gases, simultaneously offering accurate and precise experimental control and tractable theoretical models, offer an excellent platform for exploring nonequilibrium phenomena in quantum systems [17–24].

In this paper we develop a tractable theoretical model to describe a prototypical nonequilibrium quantum system: a multimode, driven-dissipative Josephson array. This system was considered in the experiment by Labouvie *et al.* [4], and is shown schematically in Fig. 1. The system consists of a long stack of "pancake-shaped" Bose-Einstein condensates (BECs), which are tightly confined in $z$ and harmonically confined in the radial ($r$) plane [Fig. 1(a)]. Atom losses at a controllable rate $\gamma$ are introduced to a single site by a tightly-focussed electron beam, and refilling of the lossy site is provided by the remainder of the lattice — an influx of particles is caused by the resulting chemical potential imbalance, with the refilling rate related to the nearest-neighbour lattice hopping $J$.

The primary result of the experiment was that within a range of dissipation rates $\gamma$ the system exhibited *bistability* — the steady-state atom number of the lossy site depended on whether it was initially full or empty. The upper (full) branch, which exhibited high particle current associated with superfluid transport from the reservoir, could be partially explained by a mean-field, tight binding model [4, 24–26], while the steady-state filling in the lower (empty) branch could be explained by an effective single particle model with decoherence and a particle-dependent driving rate [4, 11]. While the basic features of each branch could therefore be understood, a cohesive theoretical description of the system within a single model is currently lacking, and several interesting observations remain unexplained. In particular, in the lower branch of the bistability region a critical slowing down was observed, suggesting the occurrence of a nonequilibrium condensation phenomenon [6,17,27] possibly akin to exciton-polariton condensation [4, 28]. Additionally, the critical dissipation observed in the superfluid phase was considerably smaller than was predicted by the simple single-mode theory [4].

Here we develop a comprehensive model for this system within the framework of c-field theory and numerically explore its features. We develop a model to describe the dynamics of the lossy site (the system) in terms of a driven-dissipative stochastic Gross-Pitaevskii



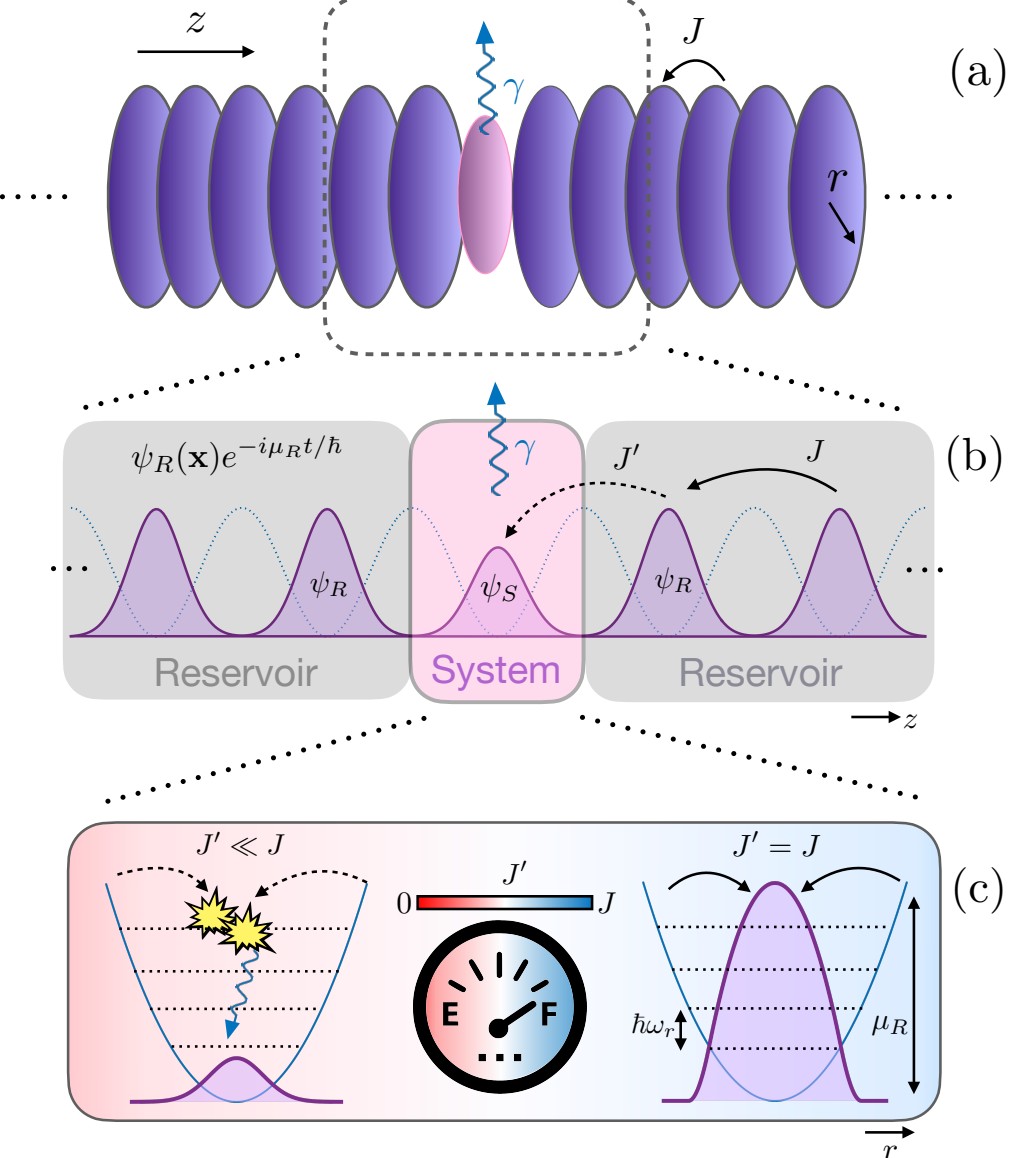

Figure 1: (a) The driven-dissipative Josephson junction array consists of a long chain of pancake-shaped condensates. Particle loss at a rate $\gamma$ is introduced to a single site by a focussed electron beam; refilling is provided from the neighbouring sites in the lattice through the nearest-neighbour hopping $J$. (b) A zoomed-in view of the chain, viewed along $z$. We develop an effective model for the dynamics of the condensate at the lossy (system) site, described by field $\psi_S(\mathbf{x}, t)$, by treating the remaining sites as a particle reservoir that provides an AC driving term of spatial profile $\psi_R(\mathbf{x})$ and frequency $\mu_R/\hbar$. (c) View of the system site in the radial direction $r$, highlighting the filling-dependent nature of the driving mechanism. The effective driving $J' = \eta J$ depends on the bare tunnelling $J$ of the neighbouring Wannier sites in $z$, but also the overlap $\eta \propto \langle \psi_S | \psi_R \rangle$ between the radial modes of the system and reservoir. The chemical potential of a full site $\mu_R$ is much larger than the radial harmonic oscillator spacing $\hbar\omega_r$; thus, for a full (**F**) site (right), particles may resonantly tunnel directly into the condensate, whereas for a nearly empty (**E**) site (left) particles instead must tunnel into an excited state and the population of the condensate may only increase via collisional relaxation.

equation, where the remainder of the lattice (the reservoir) provides an AC driving on the system [Fig. 1(b)]. The model is related to the Lugiato-Lefever equation of nonlinear optics [29, 30]. Crucially, in contrast to previous models [4], our approach explicitly includes the radial multimode dynamics of this nonequilibrium quantum system while remaining theoretically tractable. As emphasized by Labouvie *et al.* [4], the chemical potential of a full site is much larger than the energy spacing between radial harmonic oscillator modes [Fig. 1(c)], and thus radial fluctuations are critical to describing the system dynamics. These fluctuations lead to a filling-dependent driving rate $J' \leq J$. When the system site is near full, particles may resonantly couple from the reservoir to the condensate. However, when the system site is depleted, particles tunnelling from the neighbouring reservoir sites are too energetic to directly access the system condensate; they must instead tunnel into an excited radial mode resonant with their energy.

By using a "coherent reservoir model" whereby the reservoirs are described by an AC driving term, we find the c-field model is capable of capturing the bistability and the critical slowing down in the lower branch, and produces a similar nonequilibrium phase diagram to Labouvie *et al.* [4]. We then investigate a "dynamical reservoir model", incorporating both the effects of finite temperature, as well as the back-action of the system on the reservoirs. This leads to some decoherence between the system and the reservoir, and in doing so we find that the model is capable of quantitatively reproducing the phase boundaries of the nonequilibrium phase diagram observed in the experiment.

The outline of the paper is as follows. In Sec. 2 we introduce the c-field formalism for the study of the driven-dissipative site. In Sec. 3 we reduce our model to a single-mode approximation, and compare analytical results obtained against a tight-binding approach. Section 4 presents numerical simulations of a multimode coherent reservoir model, and Sec. 5 presents a multimode dynamical reservoir model that includes back-action and heating effects on the driving sites. Section 6 presents discussion and suggestions for future work, before we conclude in Sec. 7.

## 2 c-field model

A natural framework for developing a nonequilibrium multimode model of the system site is provided by c-field theory, also known as the truncated Wigner approximation (TWA) for the interacting Bose gas [31]. Briefly, a master equation for the dynamics of the low-energy, classical modes of the system can be transformed into an equation of motion for the system's Wigner function. Neglecting third-order and higher derivatives — justified at short times or when the number of particles in the system is large — leads to a Fokker-Planck equation, which can then be simulated by an ensemble of trajectories that are solutions to a stochastic differential equation. This approach has been successfully applied to the simulation of the dynamics of Bose-Einstein condensates in a wide range of circumstances [32].

In this work the system of interest is the lattice site where there are atom losses caused by a focused electron beam. The system is described by a classical field $\psi_S(\mathbf{x}, t)$. The dynamics of $\psi_S(\mathbf{x}, t)$ are governed by a damped and driven stochastic projected Gross-Pitaevskii equation (SPGPE)

$$i\hbar \, d\psi_S = \{(\mathcal{L} - i\gamma/2)\psi_S + \mathcal{F}\} \, dt + dW \,, \tag{1}$$

where $\mathcal{L}$ is the Gross-Pitaevskii operator, $\gamma$ is the particle loss rate, $\mathcal{F}$ is the driving associated with the reservoir sites, and $dW$ is a Wiener noise term associated with the losses. The projected GP operator is given by

$$\mathcal{L}\psi_S \equiv \mathcal{P}\left\{\left[-\frac{\hbar\nabla^2}{2m} + V(\mathbf{x}) + g_2|\psi_S|^2\right]\psi_S\right\},\tag{2}$$

where the external potential is a symmetric harmonic trap of the form

$$V(\mathbf{x}) = \frac{1}{2}m\omega_r^2(x^2 + y^2).\tag{3}$$

The effective 2D interaction parameter is given by $g_2 = g\int dz|w(z)|^4$, where $g = 4\pi\hbar^2 a_s/m$ for s-wave scattering length $a_s$, and $w(z)$ is the Wannier function associated with the optical lattice in the $z$-direction [33]. The projection operator $\mathcal{P}$ confines the evolution to the c-field region

$$\mathcal{P}\{f(\mathbf{x})\} = \sum_{n\in C}\varphi_n(\mathbf{x})\int d^2\mathbf{x}'\, \varphi_n^*(\mathbf{x}')f(\mathbf{x}'),\tag{4}$$

where $C$ contains all modes with energies less than a prescribed energy cutoff; here this amounts to $n_x + n_y \leq n_{\text{cut}}$, where $n_x, n_y$ are the quantum numbers of the Cartesian basis states of a symmetric 2D harmonic oscillator. The role of the projection operator is to properly restrict the classical field approach to only the highly occupied modes for which the approach is valid; the number of states required depends on the chemical potential of the system. The inclusion of a projector allows the classical field to thermalize, and thus measures such as the condensate fraction can be calculated from the field correlations. For further details on technical aspects of the projector see, e.g., [31,32].

We assume the condensate is "optically thin" in the z-dimension, i.e., that the intensity of the electron beam does not diminish along the propagation axis, such that the loss rate $\gamma$ can be treated as spatially independent. The Wiener noise term $dW$ associated with the dissipation therefore satisfies

$$\langle dW^*(\mathbf{x}, t)dW(\mathbf{x}', t')\rangle = \gamma\, \delta_C(\mathbf{x}, \mathbf{x}')\delta(t - t')dt,\tag{5}$$

where $\delta_c(\mathbf{x}, \mathbf{x}') \equiv \sum_{n\in C}\varphi_n(\mathbf{x})\varphi_n(\mathbf{x}')$ is the kernel of the coherent region projection operator [31]; it behaves as a Dirac delta function when acting on the projected subspace.

The last remaining ingredient is the forcing function $\mathcal{F}$, which physically is provided by the neighbouring sites in the lattice. The experiment of Labouvie et al. [4] spanned approximately $\sim 60$ individual sites. Instead of simulating the entire system, we begin with a simple approximation; we treat the two neighbouring sites as undepleted reservoirs at zero temperature. The driving function $\mathcal{F}$ is thus described by a field which is an equilibrium solution to the Gross-Pitaevskii equation with spatial profile $\psi_R(\mathbf{x})$ and chemical potential $\mu_R$. The forcing hence takes the form of a spatially dependent coherent AC driving term (cf. Fig. 1)

$$\mathcal{F}(\mathbf{x}, t) = -2J\, \psi_R(\mathbf{x})\, e^{-i\mu_R t/\hbar},\tag{6}$$

where the factor of two comes from the fact that the system has two nearest neighbours. In principle there could be a relative phase between the two driving sites — this would result in the replacement $2 \rightarrow (1 + e^{i\varphi})$ in Eq. (6), but at this level of approximation this would only result in reduced effective value of $|J|$. We therefore ignore this possibility for the moment, but will consider it further in Sec 5.

For an optical lattice with height $V_0$, the bare tunnelling rate $J$ is related to the matrix elements of neighbouring Wannier functions via [33]

$$J = \int dz\, w^*(z)\hat{H}w(z - a),\tag{7}$$

where $\hat{H} = -\hbar^2\partial_z^2/2m + V_0\sin^2(2\pi z/\lambda)$ and $a = \lambda/2$ is the lattice spacing. The effective rate of tunnelling however depends on the overlap between the radial wave functions of the system

and reservoir at a given instant in time. From Eq. (1) it follows that the mean system atom number $N_S$ evolves according to

$$dN_S/dt = \hbar^{-1}\left(4\eta J\sqrt{N_R N_S} - \gamma N_S\right),\tag{8}$$

where $N_R$ is the reservoir atom number, and $\eta$ is a so-called *Frank-Condon factor* given by

$$\eta(t) = \frac{\text{Im}\{\langle\psi_R|\psi_S(t)\rangle\}}{\sqrt{N_R N_S}}, \qquad |\eta| \le 1,\tag{9}$$

such that in the steady state $N_S/N_R = (4J\eta/\gamma)^2$. Note that $\eta$ is inherently built into the c-field model presented here, whereas in Labouvie *et al.* [4] this factor was added phenomenologically and assumed to be linear in the atom number difference.

The driving reservoir field $\psi_R(\mathbf{x})$ is determined by solving for the ground state of the projected GPE by integrating the damped projected GPE

$$i\hbar\partial_t\psi_R = (1 - i\gamma)(\mathcal{L} - \mu_R)\psi_R = 0.\tag{10}$$

A given value of the reservoir chemical potential $\mu_R$ hence self-consistently determines both the forcing profile $J\psi_R(\mathbf{x})$, and driving frequency $\mu_R/\hbar$.

## 2.1 Relation to Lugiato-Lefever model

The model we have presented for the driven-dissipative BEC Eq. (1) is closely related to the the Lugiato-Lefever equation (LLE) [29, 30], which is usually expressed in the form

$$\partial_t\psi = -i\alpha\nabla^2\psi - i\beta|\psi|^2\psi - (\gamma + i\Omega)\psi - \delta.\tag{11}$$

The LLE is widely used in the setting of nonlinear optics, and can also be derived in the small amplitude limit from the AC-driven damped sine-Gordon equation [34, 35]. As such the LLE can describe a variety of systems, including coupled pendula, extended Josephson junctions, easy-axis ferromagnets in microwave fields, rf-driven plasmas, whispering gallery mode resonators, and chemical reaction-diffusion systems (see, e.g., Refs. [35, 36] and references therein). Our system corresponds to the case of anomalous dispersion $\alpha < 0$, defocussing nonlinearity $\beta > 0$, and blue-detuned driving $\Omega > 0$; however, all of these may in general be of either sign [35] depending on the physical system. The LLE and generalized versions also arise in the context of superfluid exciton-polariton systems with coherent pumping [28, 37–42].

## 3 Single-mode approximation

Before solving the full SPGPE (1), let us first consider the mean-field treatment when the system consists of a single mode. We will find that this approximation provides a qualitative understanding of the bistability and hysteresis observed in Labouvie *et al.* [4]. Furthermore, analytical results can be derived for this model and these will provide a useful point for comparison with the numerical analysis of the subsequent sections.

We assume that the system is dominated by a single mode, such that $\psi_S(x,y,t) \approx c_S(t)\phi_0(x)\phi_0(y)$ and $\mathcal{L}\psi_S = \mu_S\psi_S$. In the mean-field approximation the noise term can be neglected, and Eq. (1) reduces to

$$i\hbar\frac{dc_S}{dt} = \mu_S c_S + g_0|c_S|^2 c_S - 2Jc_R - i\frac{\gamma}{2}c_S,\tag{12}$$

where $g_0 = g_2 \left( \int dx \, |\phi_0(x)|^4 \right)^2$. Eliminating the $\mu_S$ term by moving to a rotating frame, $c_S \to c_S e^{i\mu_S t/\hbar}$, the driving term takes the form

$$c_R = \sqrt{n_R} e^{-i\Delta t}, \tag{13}$$

where $\hbar\Delta = \mu_R - \mu_S \geq 0$ is the detuning, and

$$c_S = \sqrt{n_S} e^{-i(\omega t - \phi)}. \tag{14}$$

where $n_S$, $\omega$ and $\phi$ are to be determined. For a steady state solution, the system must be locked to the reservoir driving frequency, with $\omega = \Delta = $ const., giving the two conditions

$$\frac{n_S}{n_R} = \left( \frac{4J \sin\phi}{\gamma} \right)^2, \qquad \frac{n_S}{n_R} = \left( \frac{2J \cos\phi}{g_0 n_S - \hbar\Delta} \right)^2. \tag{15}$$

Eliminating $\phi$ then yields the following cubic equation for the density

$$n_S \left[ (g_0 n_S - \hbar\Delta)^2 + \gamma^2/4 \right] = 4J^2 n_R. \tag{16}$$

Real and positive roots of Eq. (16) define steady-state solutions, phase locked to the drive, with a constant phase lag $\phi$ given by

$$\tan\phi = \frac{\gamma/2}{g_0 n_S - \hbar\Delta} + \pi \Theta(\hbar\Delta - g_0 n_S). \tag{17}$$

The second term involving the Heaviside function $\Theta(x)$ arises as Eq. (15) only defines $\phi$ up to a shift of $\pi$; this ensures $\phi$ varies continuously on the interval $[0, \pi]$.

It is convenient to define $\Delta^{-1}$ and $\hbar\Delta$ as the natural units of time and energy respectively, reducing Eq. (16) to the dimensionless form

$$n_S^3 - 2n_S^2 + (1 + \tilde{\gamma}^2)n_S = \tilde{J}^2, \tag{18}$$

where $\tilde{\gamma} = \gamma/2\hbar\Delta$ and $\tilde{J} = \sqrt{n_R} 2J/\hbar\Delta$. For convenience we have also absorbed the nonlinearity coefficient $g_0/\hbar\Delta$ via the substitution $\{c_S, c_R\} \to \sqrt{g_0/\hbar\Delta} \{c_S, c_R\}$ such that $n_R = 1$.

Equation (18) permits either one or three solutions (real and positive), depending on the values of $\tilde{\gamma}$ and $\tilde{J}$ [see Fig. 2(a)]. The region where multiple solutions exist is bounded by the critical points; differentiating Eq. (18) with respect to $n_S$ yields the density at these critical points, $n_\pm$, as

$$n_\pm(\tilde{\gamma}) = \frac{2 \pm \sqrt{1 - 3\tilde{\gamma}^2}}{3}, \tag{19}$$

as graphed in Fig. 2(b). For $\tilde{\gamma}^2 > 1/3$ Eq. (19) no longer yields real solutions, indicating that there is only one solution [Fig. 2 (a), right curve]. For $\tilde{\gamma}^2 < 1/3$, multiple solutions emerge via a pitchfork bifurcation [38]; either one or three solutions exist depending on the value of $\tilde{J}$ [see Fig. 2(a), left curve]. Inserting Eq. (19) into Eq. (18) gives the range of $\tilde{J}$ that supports multiple solutions, $\tilde{J}_- \leq \tilde{J} \leq \tilde{J}_+$, where

$$\tilde{J}_\pm^2(\tilde{\gamma}) = \frac{2}{27} \left[ 1 + 9\tilde{\gamma}^2 \pm (1 - 3\tilde{\gamma}^2)^{3/2} \right], \tag{20}$$

as shown in Fig. 2(c). To assess the number of stable solutions, we write $c_S \to c_S + \delta c e^{-i\lambda t}$, with $c_S$ given by Eqs. (14), (16) and (17). Retaining terms linear in $\delta c$ yields

$$\lambda = -i\tilde{\gamma} \pm \sqrt{(n_S - 1)(3n_S - 1)}, \tag{21}$$

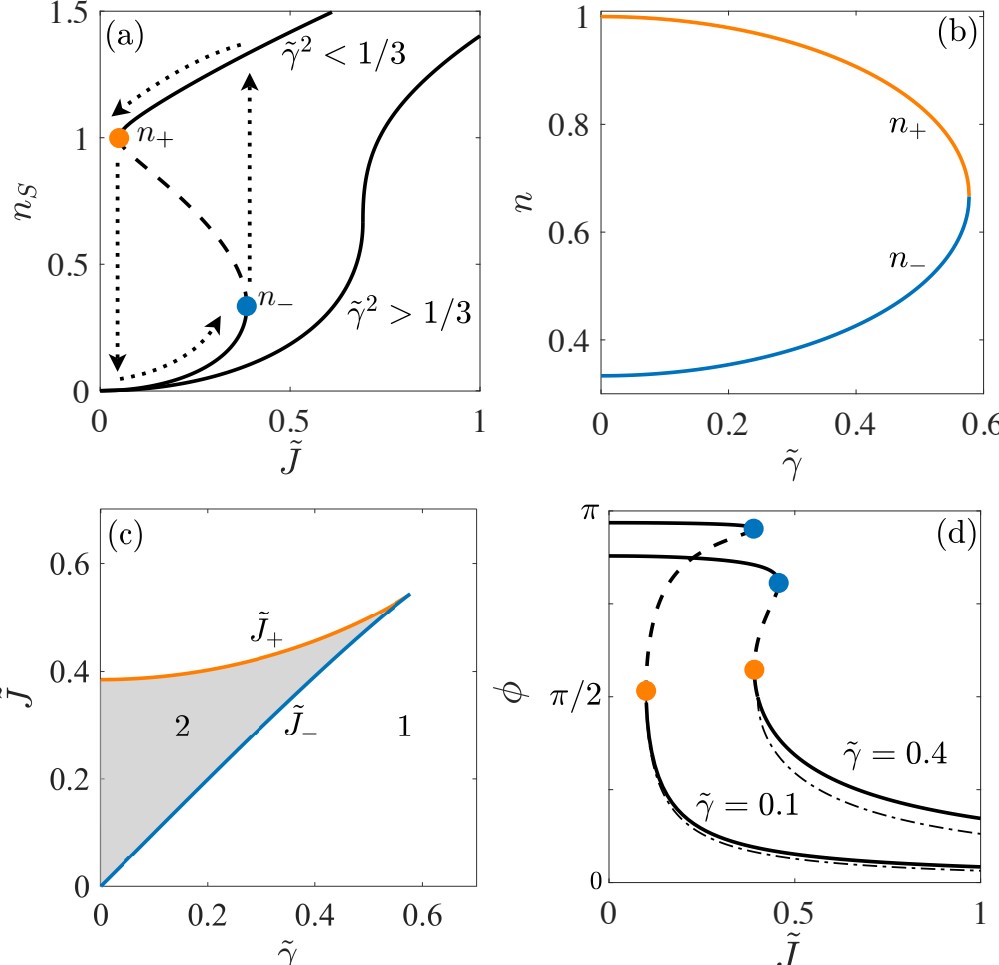

Figure 2: Bistability of the single-mode model. (a) Examples of $n_S$ vs. $\tilde{J}$ curves in the bistable ($\tilde{\gamma}^2 \leq 1/3$) and monostable regions ($\tilde{\gamma}^2 > 1/3$) of parameter space. The dashed region indicates the unstable solution, and the circle markers indicate the critical points ($\tilde{J}_-, n_+$) and ($\tilde{J}_+, n_-$). (b) Occupation number vs. $\tilde{\gamma}$ for the critical points $n_+$ and $n_-$. (c) Bistability phase diagram in the $\tilde{\gamma} - \tilde{J}$ plane. The shaded region exhibits bistability (two stable solutions) whereas the unshaded regions contains only one solution. (d) Examples of the phase lag $\phi$ against $\tilde{J}$ for large and small dissipation values where bistability is observed. As in (a), the dashed line indicates the unstable region, and circle markers show the critical points. The dash-dot lines show the Josephson model prediction for the upper branch [Eq. (24)] for comparison (see text).

giving stable solutions whenever $\text{Im}\{\lambda\} \leq 0$, i.e.,

$$\text{Im}\left\{\sqrt{(n_S - 1)(3n_S - 1)}\right\} \leq \tilde{\gamma}. \tag{22}$$

This condition is satisfied for $n_S \leq n_-$ or $n_S \geq n_+$ as per Eq. (19). The steady-state populations can hence be separated into a lower branch $n_S \leq n_-$, a middle branch $n_- < n_S < n_+$, and an upper branch $n_S \geq n_+$. The upper and lower branches are stable and the central branch is unstable, i.e., the system is bistable, and exhibits a hysteresis cycle as shown in Fig 2(a). The shaded region bounded by $J_\pm$ as shown in Fig. 2(c) thus defines the bistable region of the $\gamma$-$J$ plane.

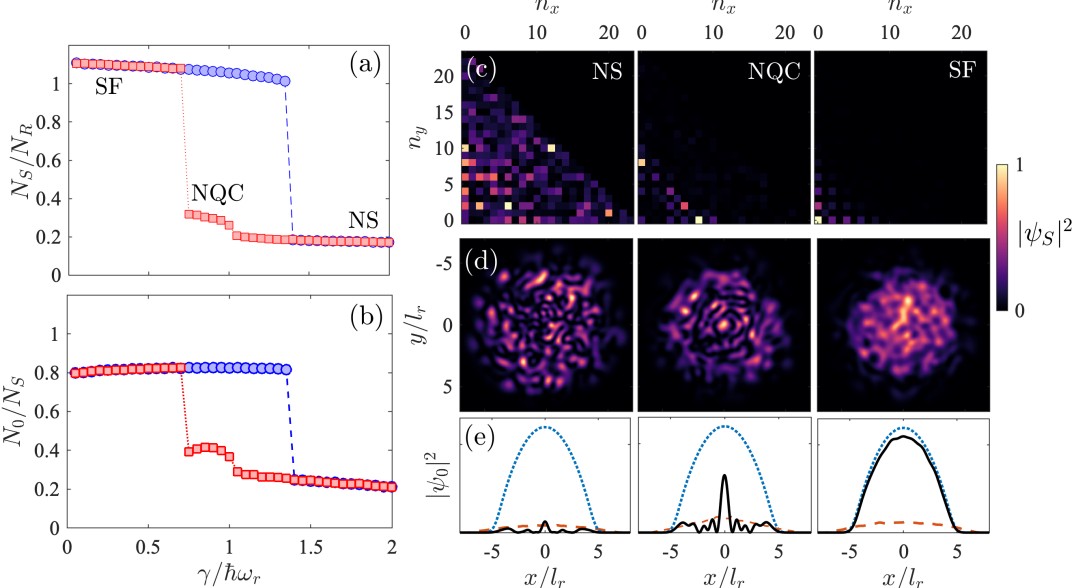

Figure 3: (a) Relative filling $N_S/N_R$, and (b) condensate fraction $N_0/N_S$ vs. dissipation $\gamma$ for at a fixed driving strength $J/\hbar\omega_r = 0.35$. Blue circle and red square markers indicate the resulting steady state from beginning with full and empty initial conditions respectively (see text). (c,d): Examples of the instantaneous particle density $|\psi_S|^2$ for the normal state (NS, $\gamma = 1.8\hbar\omega_r$), nonequilibrium quasicondensate (NQC, $\gamma = 0.7\hbar\omega_r$) and superfluid (SF, $\gamma = 0.65\hbar\omega_r$) phases, in (c) mode space, and (d) position space. (e) Slices along the $x$-axis at $y = 0$, showing the condensate density $|\psi_0|^2$ (black solid), final time-averaged noncondensate density (red dashed), and reservoir density (blue dotted).

Mathematically the above results are identical to those obtained for a single mode dispersive optical cavity with cubic nonlinearity [38, 43]. Although in the above analysis we have neglected the effects of noise, the single mode system can in fact be solved exactly, with the inclusion of fluctuations, within the generalized P representation [43]. As discussed in Refs [38, 43], retaining fluctuations in the single-mode model does not dramatically change the mean-field stability diagram; rather the essential difference is that steady-state filling becomes *bimodal* rather than truly bistable — the system tends to explore the vicinity near the stable mean field solutions on shorter timescales, and stochastically switches between the two at longer timescales. The inclusion of noise therefore does not drastically alter the qualitative behaviour from the mean-field predictions, suggesting that multimode effects are essential to capture the system behaviour observed in the experiment of Labouvie *et al.* [4].

As Eq. (12) is a rather drastic simplification of the system under consideration, it is useful to compare the above results with the Josephson model considered in Labouvie *et al.* [4]. This approach instead considers an infinite array of coupled, single-mode sites

$$i\hbar\frac{dc_n}{dt} = -J(c_{n+1} + c_{n-1}) + g_0|c_n|^2 c_n - i\frac{\gamma}{2}c_n\delta_{mn}. \qquad (23)$$

In addition to the trivial steady state $c_n = 0 \;\forall n$, this model supports a steady solution in which all sites have identical filling, and a relative phase difference $\phi \equiv |\phi_n - \phi_{n+1}|$ between neighbouring sites given by

$$\sin\phi = \frac{\gamma}{4J}. \qquad (24)$$

Clearly, the Josephson model predicts a breakdown of superflow at the critical value $\gamma_c = 4J$,

since Eq. (24) has no solution for $\gamma > \gamma_c$.

Whereas the Josephson model only supports identical filling across all sites, the solutions to Eq. (16) can exhibit an "overfilling" ($n_S/n_R > 1$) under strong driving [Fig. 2(a)]. This overfilling is unphysical in the context of the driven damped lattice system we consider here (no such behaviour was observed in Labouvie *et al.* [4]) and, as we will show in Sec. 5, is a consequence of neglecting the back-action of the system on the reservoir. However, here we are primarily interested in the bistability region, where this effect is less pronounced than for large values of the driving. Further, comparing Eqs. (15) and (24) shows the phase lag differs by a factor of $\sqrt{n_S/n_R}$. This discrepancy also turns out to be small in the vicinity of the bistability region; the predicted phase lag curves are in semi-quantitative agreement, especially for smaller values of dissipation, see Fig. 2(d).

Finally, note that both models yield very similar predictions for the critical dissipation strength of the superfluid branch: expanding Eq. (20) to first order in $\tilde{\gamma}$ yields $\tilde{J}_- \approx \tilde{\gamma}$, which is equivalent to $\gamma_c = 4J$ as predicted in the Josephson model – see Fig 2(c).

From the preceeding analysis, it is clear that the driven-damped single mode model captures the qualitative behaviour of the bistability. However, it does not capture many of the other key findings of Labouvie *et al.* [4]. In particular, the bistability region boundaries differ in shape to those observed in the experiment, and further the solutions of Eq. (12) show no indication of critical slowing down within the bistable region. This indicates the radial fluctuations and other sources of noise are important for capturing the experimental observations. We address this in the following section, where we turn to numerically solving the full multimode model.

## 4  Multimode coherent reservoir model

### 4.1  Numerical implementation

We now proceed with a full numerical treatment of the driven damped SPGPE, which includes both radial fluctuations of the condensate as well as noise associated with the dissipation mechanism [see Eq. (1)]. Following the standard procedure for c-field implementations, Eq. (1) is numerically integrated via a pseudospectral method with quadrature (formally equivalent to the Galerkin method [31, 44]). The basis is a standard Cartesian Hermite-Gauss representation, which diagonalizes the single-particle Hamiltonian and thus approximately diagonalizes the nonlinear problem when the mode occupation is low [45]. The nonlinear term is treated using the appropriate Gaussian quadrature rules [31], such that the matrix elements are implemented exactly for all modes in the c-field region. The simulations were performed using XMDS2 [46]. Throughout we work in units of the radial harmonic oscillator, i.e., length, time and energy are expressed in units of $l_r = \sqrt{\hbar/m\omega_r}$, $\omega_r^{-1}$ and $\hbar\omega_r$ respectively.

As usual with c-field implementations, some quantities (such as total energy and particle number) inevitably exhibit a dependence on the chosen energy cutoff. Some care is therefore required in both choosing the energy cutoff, and interpretation of results. To verify that our results did not depend on the choice of cutoff, we calculated the condensate number $N_0$, as determined from the Onsager-Penrose criterion [31], which specifies the condensate number as the largest eigenvalue of the one-body density matrix $\rho(\mathbf{r}, \mathbf{r}')$. Within our c-field framework this is readily constructed from a time-average of the steady state as

$$\rho(\mathbf{r}, \mathbf{r}') = N_t^{-1} \sum_{n=1}^{N_t} \psi_S^*(\mathbf{r}, t_n) \psi_S(\mathbf{r}', t_n). \tag{25}$$

Varying the cutoff in the range of $\sim 2\mu_R$ to $3\mu_R$, we found that $N_0$ was insensitive to the

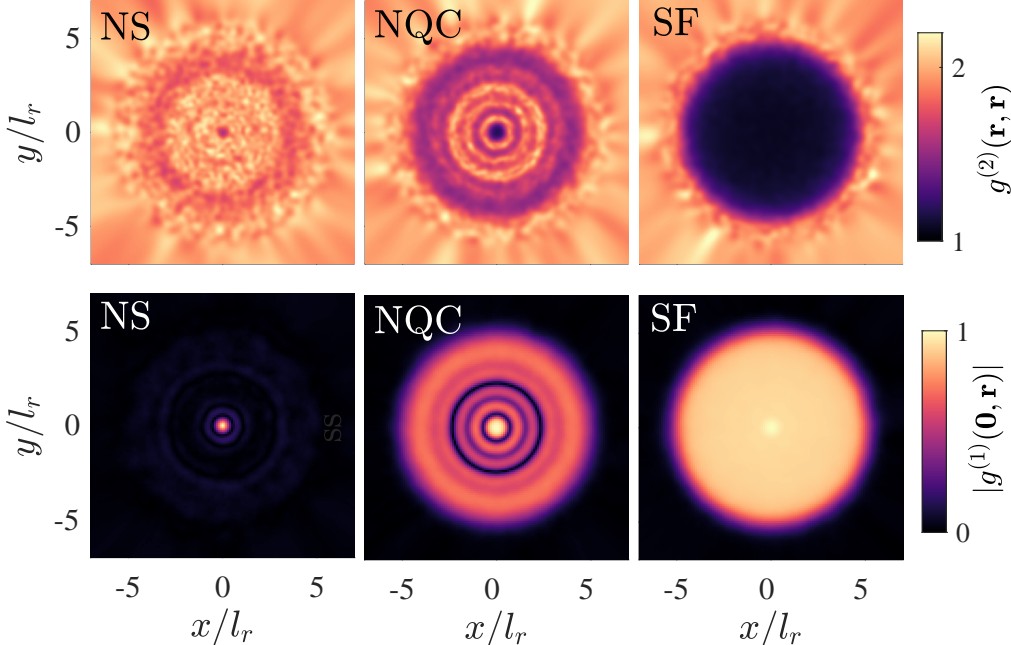

Figure 4: First- and second-order correlation functions $g^{(2)}(\mathbf{r},\mathbf{r})$ (top row) and $g^{(1)}(\mathbf{0},\mathbf{r})$ (bottom row) for the three phases shown in Fig. 3. Left column: normal state. Middle column: nonequilibrium quasicondensate . Right column: superfluid.

choice in cutoff (variations were on the order of 5%). The values for $\gamma$ and $J$ at which we observed the key qualitative changes in the system behaviour were also not sensitive to the choice of cutoff, and nor were the qualitative trends observed in the particle number and condensate fraction. Our results for the nonequilibrium phase boundaries can thus be viewed as quantitative predictions, whereas the condensate fraction and particle number only indicate qualitative trends. Unless otherwise specified, we used the dimensionless interaction parameter $C_{nl} = g_2/\hbar\omega_r l_r^2 = 0.2$ and $\mu_R/\hbar\omega_r = 12$, giving $N_R \approx 2200$ atoms in the driving reservoir. The cutoff in mode space is set to $n_{\text{cut}} = 2\mu_R/\hbar\omega_r (= 24)$.

## 4.2 Nonequilibrium steady states

As a driven-dissipative nonlinear system, the steady states of the model are in general expected to be dependent on the initial conditions. Following Labouvie *et al.* [4], we therefore compare the results of beginning with an initially phase coherent "full" site, against an initially incoherent "empty" site. For the full site the initial condition is the same as the reservoir wave function, $\psi_S(\mathbf{x}, t = 0) = \psi_R(\mathbf{x})$, whereas the empty site has each single particle mode seeded with complex Gaussian noise, scaled such that the initial relative filling $N_S/N_R$ is 5%. Figure 3 shows the steady state properties of these two cases as $\gamma$ is varied, for a fixed driving strength of $J/\hbar\omega_r = 0.35$.

Despite the additional features of multimode collisions, and the driving noise associated with the atom loss, the multimode coherent reservoir model results for the full branch [blue circles] show little difference from those of the single-mode model discussed in Sec. 3. Below the expected dissipation threshold $\gamma_c \sim 4J$, the system is able to remain in the superfluid phase (SF), and retains approximately unit relative filling [Fig. 3(a)] and condensate fraction $N_0/N_S \sim 80\%$ [Fig. 3(b)]. For dissipation above the expected threshold the system eventually becomes depleted, and the steady state phase is an incoherent 'normal state' (NS), containing only a residual relative filling and condensate fraction of $\sim 20\%$ [Fig. 3(a,b)]. The reduc-

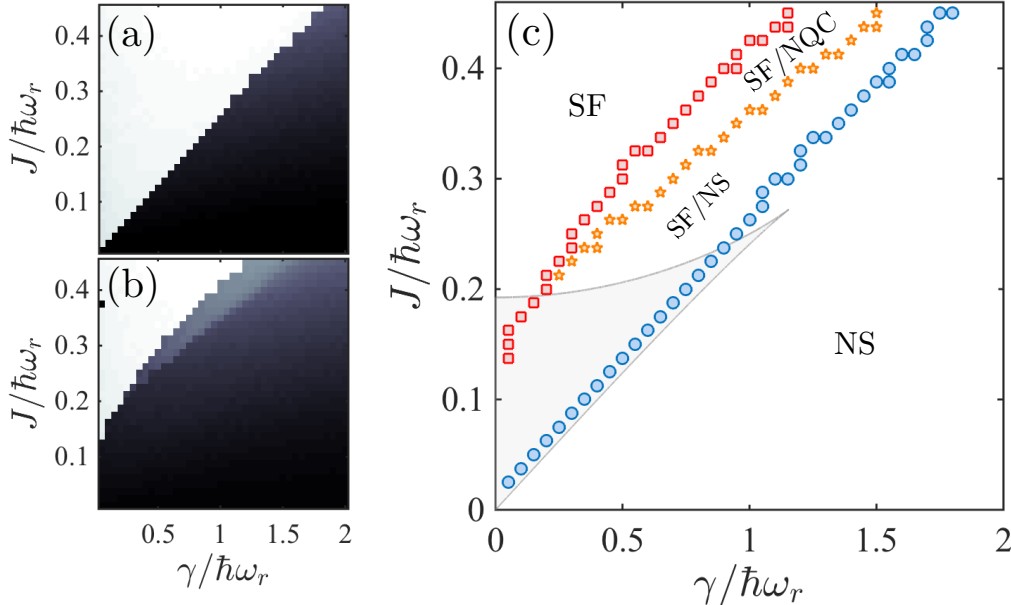

Figure 5: Steady-state condensate fractions $N_0/N_S$ in the (a) full and (b) empty branches in the $J$-$\gamma$ plane. (c) The phase diagram determined by numerically extracting phase boundaries from (a) and (b). The upper and lower boundaries were extracted by detecting the boundary where the condensate fraction exceeded 0.7, and the middle boundary shows where the it exceeded 0.4. For comparison the shaded region shows the bistable region of the single-mode system [c.f. Fig. 2(c)].

tion in condensate fraction in both cases occurs due to the driving noise associated with the dissipation.

In the normal state the particle density is dominated by thermal-like fluctuations, as can be seen in the instantaneous density profiles, [Fig. 3(c,d), left column], and the condensate and noncondensate fraction profiles in [Fig. 3(e), left]. In contrast, in the superfluid state the density closely resembles the profile of the reservoir condensate [Fig. 3(c,d, right)]. The the condensate mode profile [Fig. 3(e, right)] also closely resembles the (equilibrium) reservoir profile, but is depleted by a small noncondensate fraction.

The "empty" branch [Fig. 3(a,b), red squares] however exhibits a feature not seen in the single-mode model; the filling and condensate fraction now exhibit an intermediate plateau in the middle of the bistable region (notice this plateau is not observed in the full branch [blue circles]). This region exhibits $\sim 30\%$ relative filling [Fig. 3(a)], $\sim 40\%$ condensate fraction [Fig. 3(b)] and reduced fluctuations in the density profile [Fig. 3(d, middle) and movie in the Supplemental Material [47]]. In this "nonequilibrium quasicondensate" phase (NQC) the majority of the particles occupy a band of harmonic oscillator modes that are resonant with the driving frequency [Fig. 3(c, middle)]. This same band of states determines the condensate mode, leading to the patterned condensate mode profile in Fig 3(e, middle). For the case $\mu_R/\hbar\omega_r = 12$ shown here, the majority of the particles occupy the band of single-particle modes satisfying $n_x + n_y \sim 8$. The populations of these modes gradually increase as $\gamma$ decreases (or $J$ increases), until they dominate over the incoherent background of the normal state. This band of excited states are preferentially excited by the AC driving due to the filling-dependent driving mechanism [see Fig. 1]; at low filling, they are near-resonant with the driving frequency $\mu$. The detuning from $\mu_R$ is a combination of the zero-point energy $\hbar\omega_R$ and a blueshift due to the repulsive interactions. States with significantly larger or smaller energies are off-resonant and hence less influenced by the driving.

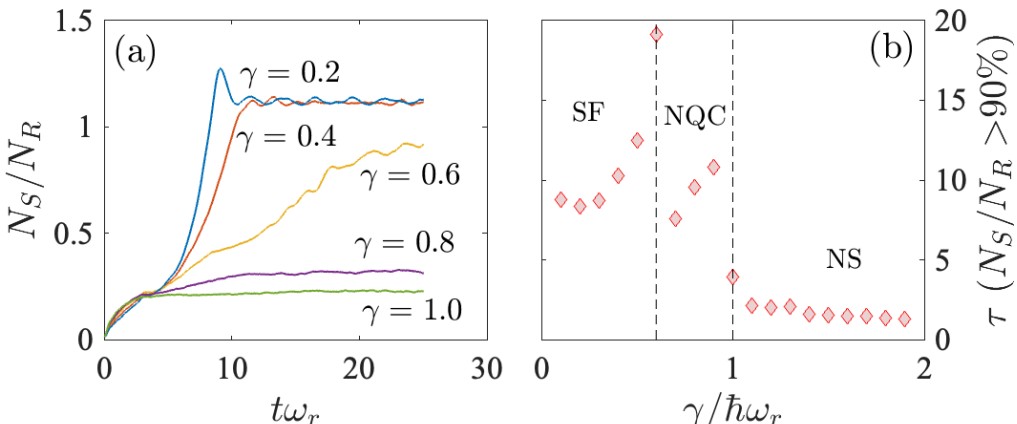

Figure 6: Critical slowing down observed at short timescales, (i.e., $t\omega_r \sim 10$, similar to the (millisecond) timescales to Labouvie *et al.* [4]). Curves were calculated from 40 trajectories for each $\gamma$. (a) Filling vs. time at fixed $J/\hbar\omega_r = 0.35$ for a range of $\gamma$. (b) Refilling time $\tau$, measured as the time taken for the relative filling to exceed 90% within the empty branch.

To gain further insight into the three nonequilibrium steady-states, in Fig. 4 we measure the spatial phase coherence via the first-order correlation function

$$g^{(1)}(\mathbf{r}, \mathbf{r}') = \frac{\langle \psi_S^*(\mathbf{r}, t)\psi_S(\mathbf{r}', t)\rangle}{[\langle |\psi_S(\mathbf{r}, t)|^2\rangle \langle |\psi_S(\mathbf{r}', t)|^2\rangle]^{1/2}}, \tag{26}$$

and the density coherence via the second-order correlation function

$$g^{(2)}(\mathbf{r}, \mathbf{r}') = \frac{\langle \psi_S^*(\mathbf{r}, t)\psi_S^*(\mathbf{r}', t)\psi_S(\mathbf{r}, t)\psi_S(\mathbf{r}', t)\rangle}{\langle |\psi_S(\mathbf{r}, t)|^2\rangle \langle |\psi_S(\mathbf{r}', t)|^2\rangle}, \tag{27}$$

where the averages are computed as time averages in the steady state as in Eq. (25). As can be seen in Fig. 4, the correlations in the normal phase and the superfluid phase of the nonequilibrium steady-states are similar to that expected at thermal equilibrium. The normal state almost completely lacks phase coherence [rapid decay of $g^{(1)}(\mathbf{0}, \mathbf{r})$], and $g^{(2)}(\mathbf{r}, \mathbf{r}) \sim 2$ throughout the system, as is typical for a thermal state [48]. Similarly the superfluid phase exhibits almost complete phase and density coherence throughout the system with $g^{(1)}(\mathbf{0}, \mathbf{r}) \sim 1$, and $g^{(2)}(\mathbf{r}, \mathbf{r}) \sim 1$, as is the case for a condensate well below the critical temperature [48].

The nonequilibrium quasicondensate phase is less typical, and exhibits a partial phase and density correlation; density and phase fluctuations are strong at intermediate radii, but less pronounced near the origin and near the edge of the cloud. The reduced coherence at intermediate radii appears to be due to a proliferation of quantized vortices, as can be seen in Fig. 3(d, middle) and the movies provided of the dynamics of filling in the three phases in the supplemental material [47].

An overview of the system's steady state behaviour is presented in Fig. 5, which shows the condensate fractions for the full and empty branches within the $\gamma$–$J$ plane [Fig. 5(a) and (b) respectively], and the corresponding nonequilibrium phase diagram extracted from the boundaries [Fig. 5(c)]. The nonequilbrium phase diagram in Fig. 5 is divided into four regions: in the top left region, the steady state of the system is the superfluid state (SF), regardless of the initial conditions, while in the bottom right the system ends up in the incoherent normal state (NS). The nonequilibrium quasicondensate boundary (orange stars) further separates the bistability region into two distinct regions; above the boundary, the bistability is between the

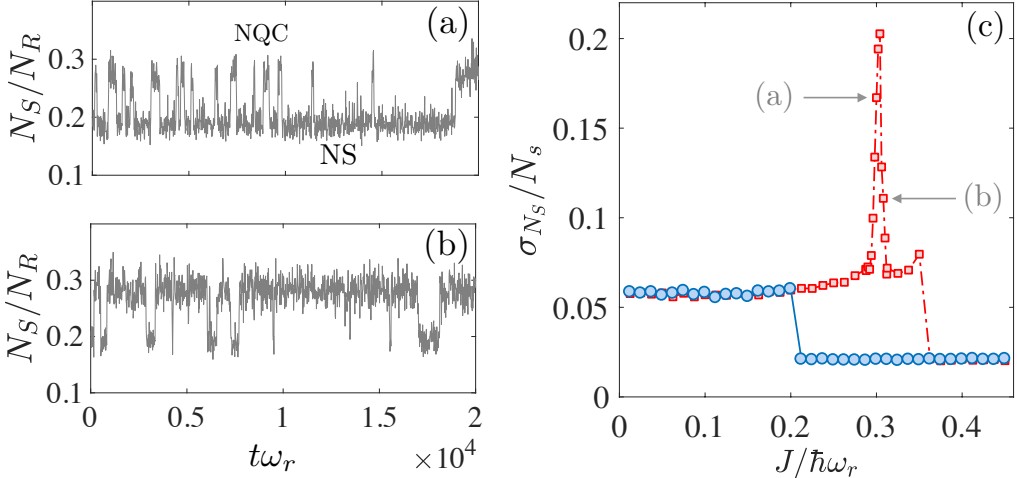

Figure 7: Critical slowing down on long timescales ($t\omega_r \gg 10$), due to atom number fluctuations in the steady state. (a,b) Examples of the filling as a function of time for driving just below and above the transition. (c) Standard deviation of the filling number as a fraction of the mean filling number, $\sigma_{N_S}/N_S$ vs. $J$, for $\gamma/\hbar\omega_r = 0.75$. Blue circles: full initial conditions, red squares: empty initial conditions. The corresponding values of $J$ for (a) and (b) are also indicated.

superfluid and nonequilibrium quasicondensate phases (SF/NQC), while below it is between the superfluid and normal phases (SF/NS).

As in the single-mode model, for an initially full site the phase transition boundary is linear in the $J$-$\gamma$ plane [Fig. 5(a,c)]. The boundary is practically indistinguishable from the Josephson model prediction $J_c = \gamma/4$, with a linear line of best fit $J_c = m\gamma + b$ yielding $m = 0.248 \pm 0.003$ and $b = 0.015$.

The changes to the lower branch, however, have greatly enlarged the bistability window [Fig 2(c)]; for most values of $\gamma$, the driving $J$ required to enter the superfluid phase is significantly higher than the single-mode prediction, and the upper boundary has changed from a convex shape to slightly concave. Notice that unlike the single mode prediction, phase boundaries no longer "close", at least over the range of $J$ and $\gamma$ considered. While these boundaries may merge at higher values of $J$ and $\gamma$, such values would not likely be accessible in a cold atom experiment and were therefore not considered in this work.

The nonequilibrium quasicondensate phase boundary, which traverses diagonally across the bistability region, only appears at sufficiently large $\mu_R$ — for example, this state was observed for $\mu_R/\hbar\omega_r = 7$ but not for $\mu_R/\hbar\omega_r \sim 3$. For $\mu_R/\hbar\omega_r \sim 3$ the bistability region quite closely resembled the predictions of the single-mode approximation (gray shaded region in [Fig 2 (c)]). In contrast to the other two boundaries, which are sharp transitions, the nonequilibrium quasicondensate transition is a smooth crossover, which becomes broader at larger $\gamma$.

## 4.3 Critical slowing down

In Labouvie *et al.* [4] a critical slowing down was observed within the bistability region, which suggestively coincides with our observation of the nonequilibrium quasicondensate phase. In Fig. 6(a) we show the dynamics of the relative filling at experimentally relevant timescales $t\omega_r \sim 10$, along with the timescale of the filling in Fig. 6(b) (defined to be when the relative filling exceeds 90%). We instead observe evidence for slowing down at both phase boundaries neighboring the bistable region. However, the observed timescales of the slowing down are broadly consistent with Labouvie *et al.* [4], as is the qualitative shape of the refilling time vs. $\gamma$.

An additional signature of critical slowing down at the nonequilibrium quasicondensate transition occurs on much longer timescales; here large atom number fluctuations are observed, as shown in Fig 7. As in the quantum theory of optical bistability [38, 43], the noise allows for a stochastic switching between the normal and nonequilibrium quasicondensate phases within the vicinity of the transition [Fig. 7(a,b)], resulting in a clear spike in the fluctuations near the transition [Fig. 7(c)]. At larger $\gamma$ where the transition broadens, the fluctuation peak becomes broader and less pronounced. While the noise is capable of switching the system between the normal and nonequilibrium quasicondensate phases, which are relatively "nearby" in terms of atom number, the noise is not large enough to cause switching between the normal and superfluid phase, and therefore no such peak is observed at the other transition points. No indication of switching between normal and superfluid phases was observed in any of the simulations, for integration timescales up to $t\omega_r \sim 10^4$.

## 5 Multimode dynamical reservoir model

In the previous section we found that the full c-field treatment of the empty branch exhibits several differences from the single-mode model, while the full branch exhibits little difference. In particular, the multimode coherent reservoir model does not resolve the discrepancy between the normal-superfluid boundary as predicted by the Josephson model, $\gamma_c = 4J$, with that observed in the experiment of Labouvie *et al.* of $\gamma_c \approx J$ [4]. This suggests that additional physics needs to be incorporated into the model. One key approximation of the previous section was that the neighbouring reservoir sites were undepletable and at zero temperature. In this section we extend the multimode coherent reservoir model to incorporate the effects of back-action of the system site on the reservoir sites, as well as the effects of finite temperature.

A schematic of the multimode dynamical reservoir model is shown in Fig. 8. In this model we simulate the dynamics of the c-field for the condensates on the left and right sides of the system site, $\psi_L$ and $\psi_R$ respectively. Their dynamics are coupled $\psi_S$ to incorporate the backaction of the system site.

The dynamics of the system site are still governed by Eq. (1), but the forcing now takes the form

$$\mathcal{F}(\mathbf{x}, t) = -J\left[\psi_L(\mathbf{x}, t) + \psi_R(\mathbf{x}, t)\right], \tag{28}$$

which replaces Eq. (6). The refilling of the dynamical reservoir condensates by the rest of the lattice is mimicked by coupling $\psi_L$ and $\psi_R$ to a thermal bath [see Fig. 8]. The equation of motion for the reservoir sites is thus taken to be the simple-growth SPGPE [48] with the addition of back-action coupling from the system field $\psi_S$:

$$i\hbar d\psi_j = \left\{(1 - i\Gamma)(\mathcal{L}[\psi_j] - \mu_0)\psi_j - J\psi_S\right\}dt + d\zeta_j. \tag{29}$$

Here $j = \{L, R\}$ and the noise correlations are

$$\langle d\zeta_j(\mathbf{x}, t)\, d\zeta_j^*(\mathbf{x}', t')\rangle = (2\Gamma k_B T_{\text{eff}})\delta_c(\mathbf{x}, \mathbf{x}')\delta(t - t')dt, \tag{30}$$

where $T_{\text{eff}}$ is an effective temperature, $\mu_0$ is the chemical potential of the reservoirs and $k_B$ is Boltzmann's constant. We emphasise that the temperature $T_{\text{eff}}$ is not a physical temperature, but rather an *effective* temperature which, in essence, we use to collectively represent all sources of classical noise in the system. The introduction of the effective temperature parameter $T_{\text{eff}}$ serves to model not only the effects of reduced coherence due to a (thermal) noncondensate fraction, but also, e.g., the resulting dephasing of different sites in the lattice (which effectively lowers the coherence of the driving). For our purposes, the effective temperature can essentially be viewed as a phenomenological fitting parameter. Note that as we

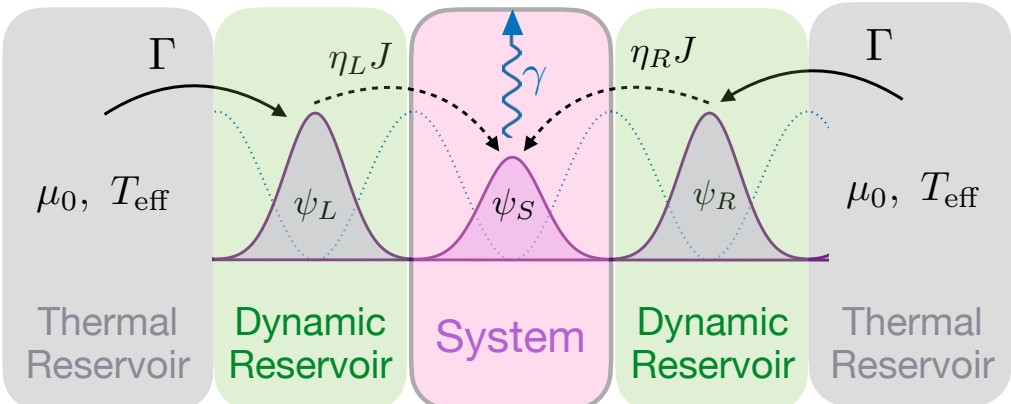

Figure 8: A schematic of the multimode dynamical reservoir model, governed by Eq. (1), and Eqs. (28)–(30). The system is now driven by two independent dynamical reservoir condensates described by $\psi_L(\mathbf{x}, t)$ and $\psi_R(\mathbf{x}, t)$, whose dynamics are governed by Eq. (29). The refilling of the dynamical reservoir condensates from the rest of the lattice is mimicked by coupling them to a thermal bath at effective temperature $T_{\text{eff}}$ and chemical potential $\mu_0$.

are using a c-field model with an energy basis cutoff, the essential observable of interest is the resulting reduced condensate fraction of the system and reservoirs, rather than the value of the temperature itself.

Under Eq. (29), the atom number on the driving sites evolves according to

$$\frac{dN_j}{dt} = \frac{2}{\hbar} \left\{ \Gamma \left[ \mu_0 - \bar{\mu}_j(t) \right] N_j - J \eta_j \sqrt{N_j N_S} \right\}, \tag{31}$$

where $\eta$ is the Franck-Condon factor from Eq. (9), and

$$\bar{\mu}_j(t) = \frac{1}{N_j} \int \mathrm{d}^2\mathbf{x} \left( \psi_j^* \mathcal{L} \psi_j + \psi_j \mathcal{L} \psi_j^* \right), \tag{32}$$

The quantity $\bar{\mu}$ may be loosely interpreted as an "instantaneous chemical potential", although strictly the chemical potential is only defined in equilibrium. The parameter $\Gamma$ thus accounts for the rate of refilling of the reservoir sites, which physically would come from the tunnelling from neighbouring lattice sites. The refilling rate depends on the instantaneous atom number and deviations of the system from its zero temperature equilibrium through the term $[\mu_0 - \bar{\mu}(t)]$. While it is therefore not entirely straightforward to determine the most appropriate choice for $\Gamma$, it is clear that it should be the same order of magnitude as $J/\hbar\omega_r$ to mimic the refilling from the rest of the lattice. For the simulation results that we present we have chosen $\Gamma = J/2\hbar\omega_r$. We found that the results did not depend on $\Gamma$ provided it was proportional to $J$ and was of a comparable magnitude.

In our simulations in this section we keep the reservoir chemical potential at $\mu_R = 12\hbar\omega_r$ as in the previous section, but increase the effective temperature. This fixes the total number of atoms at the reservoir sites, while decreasing the condensed fraction. Figure 9 shows filling measures for the multimode dynamical reservoir model as a function of $\gamma$ at fixed $J$ with an effective temperature of $k_B T_{\text{eff}}/\hbar\omega_r = 30$. This results in a condensate fraction of approximately 45% in the reservoirs when $J = 0$. Figure 9(a) shows that the addition of back-action has indeed reduced the robustness of the superfluid phase, with the collapse now occurring at $\gamma_c \approx J = 0.3$.

Further, the incorporation of a dynamical model for the reservoir sites has alleviated the unphysical "overfilling" previously observed for $\gamma \ll J$ [cf. Fig. 3(a)]; $N_S/N_R$ now only marginally

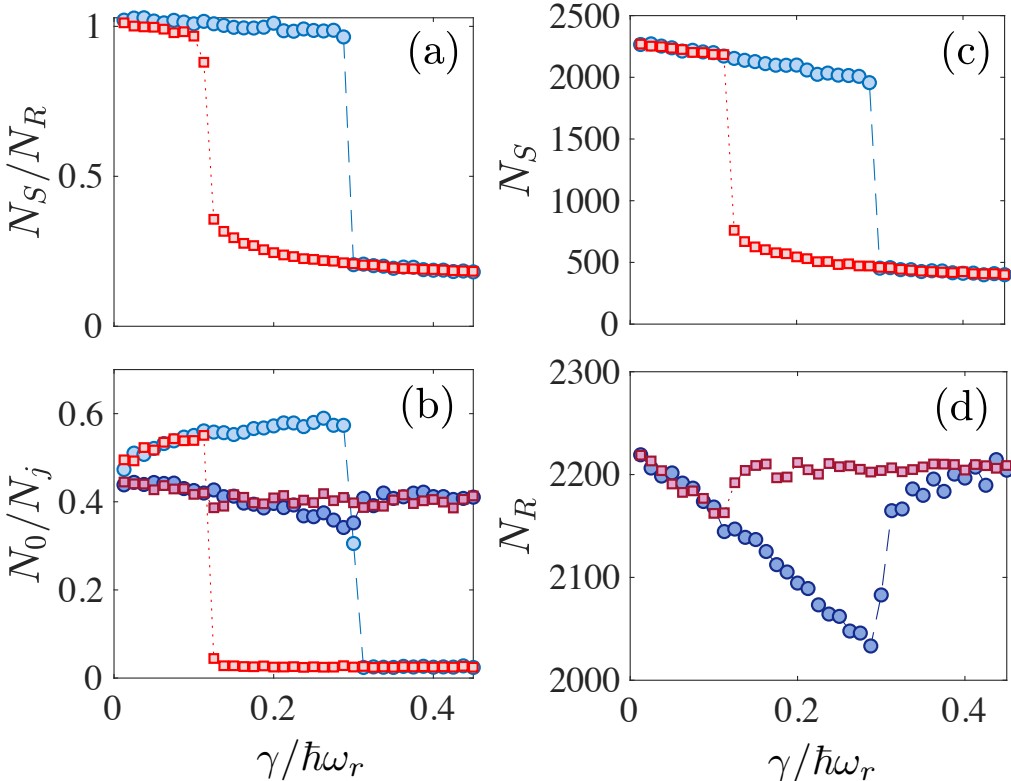

Figure 9: Filling measures vs. $\gamma$ for the dynamical reservoir driving [Eqs. (28) and (29)], for parameters $J/\hbar\omega_r = 0.3$ and $k_B T_{\text{eff}}/\hbar\omega_r = 30$. (a) Relative filling for the lower branch (red squares) and upper branch (blue circles). (b) Condensate fraction of the system site (light markers) and one of the reservoirs (dark markers) for the lower branch (squares) and the upper branch (circles). (c,d) Atom number of the system (c) and one of the reservoirs (d) [markers are as in (b)].

exceeds unity at small dissipation. In the multimode dynamical reservoir model, the nonequilibrium quasicondensate phase appears only as a transient state at intermediate times, collapsing to the normal state at later times. The plateau associated with the nonequilibrium quasicondensate phase is therefore absent in the relative filling curve, which is determined from the steady-state behaviour. However, the intermediate time window where the nonequilibrium quasicondensate is observed is similar to the experimentally relevant timescales $\sim 10 - 30$ ms, suggesting this phase is potentially experimentally relevant even though it is no longer stable at long times.

An interesting feature of Fig. 9(b) is that when the system is in the superfluid phase, the condensate fraction of the system site actually *increases* with increasing $\gamma$, at the expense of the condensate fraction of the reservoirs. A possible explanation for this outcome is that the coherent condensate atoms tunnel more easily into the system from the reservoirs than the thermal atoms thus reducing the coherence of the reservoirs and increasing that of the system. We note that similar behaviour is well known in the study of superfluid helium through capillaries, wherein the superfluid can easily flow but the normal fluid cannot. Once the system collapses into the normal phase, the filling and condensate fractions of the reservoirs recover to near their equilibrium values.

The first- and second order correlation functions for the multimode dynamical reservoir model were qualitatively unchanged from those presented for the multimode coherent reser-

voir model in Fig. 4. However, the bistability phase diagram resulting from this model has some significant differences, As shown in Fig. 10, for the choice of $k_B T_{\text{eff}}/\hbar\omega_r = 30$ the transition boundary of the upper branch is now in quantitative agreement with Labouvie *et al.* [4], with $\gamma_c \approx J$. The boundary of the lower branch is also quantitatively consistent with the observations of Labouvie *et al.* [4], being well-described by a power law fit with an exponent close to 1/2 as expected for an incoherent hopping process. We find that the effective temperature parameter essentially controls the *slope* of the  linear transition boundary, as depicted in the inset of Fig. 10(c). Thus it seems that finite temperature and dephasing effects are  likely responsible for the observation of $\gamma_c \approx J$ by Labouvie *et al.* [4]. For completeness, in Fig. 10 we also show the points where critical slowing down was observed in the experiment [orange stars], although no signature of this boundary appears in our dynamical reservoir model.

## 6  Discussion

### 6.1  Multimode coherent reservoir model

In the multimode coherent reservoir model, we found that the superfluid branch exhibits little difference from the Josephson and driven-damped single-mode models. This is perhaps not surprising, as the reservoirs are perfectly coherent and determined by the ground state of $\mathcal{L}$. Although in terms of the harmonic oscillator basis many modes are relevant, the system can still be reduced to a single mode in terms of the stationary states of the nonlinear GPE operator $\mathcal{L}$.

The appearance of the nonequilibrium quasicondensate in the lower branch, and its absence in the upper branch can be explained as follows. If the system site is full, the nonlinear interactions shift the energy such that particles from the reservoir sites can directly tunnel to the ground state of the system. However, when the system is empty the particles from the reservoir sites instead must tunnel into excited modes consistent with their energy. If the driving and dissipation occur on timescales comparable to the collision rate, then the system can thermalise and restore the condensate in the lowest-lying modes. However, if driving and dissipation operate on faster timescales than the collision rate, particles cannot reach the ground state before they are lost from the system.

The nonequilibrium phase diagram obtained from the multimode coherent reservoir model has a remarkable qualitative resemblance to that obtained in the experiments of Labouvie *et al.* [4]. The partial condensation of this phase [Fig. 3(b)] is also consistent with their reported observation of the enhanced current in this region of the phase diagram, which suggested a partial superfluid transport. However, the distinct plateau in the filling fraction seen in Fig. 3(b) was not observed in the experiment. Finally, the critical slowing down on short timescales in the multimode dynamical reservoir model (Fig. 6), which occur on timescales $\omega_r t \sim 10$, are broadly consistent with Labouvie *et al.* [4], where the timescale for slowing down was on the order of $\sim 10$ ms to $\sim 20$ ms and the trap frequency was $\omega_r = 2\pi \times 165$ Hz. The general form of the filling timescale in Fig. 6(b) also bears a clear qualitative resemblance to that observed in Labouvie *et al.* [4]. However, we note in our data that the largest timescale is between the nonequilibrium quasicondensate and normal state phases, whereas the results of Labouvie *et al.* [4] seemingly suggest the largest timescale should be between the condensate and nonequilibrium quasicondensate phases. The critical slowing down at long timescales (Fig. 7) presents a clear signature in the atom number fluctuations. Observation of this signature within a cold atom experiment may be more challenging, requiring several seconds of evolution time, but should nonetheless be within reach experimentally.

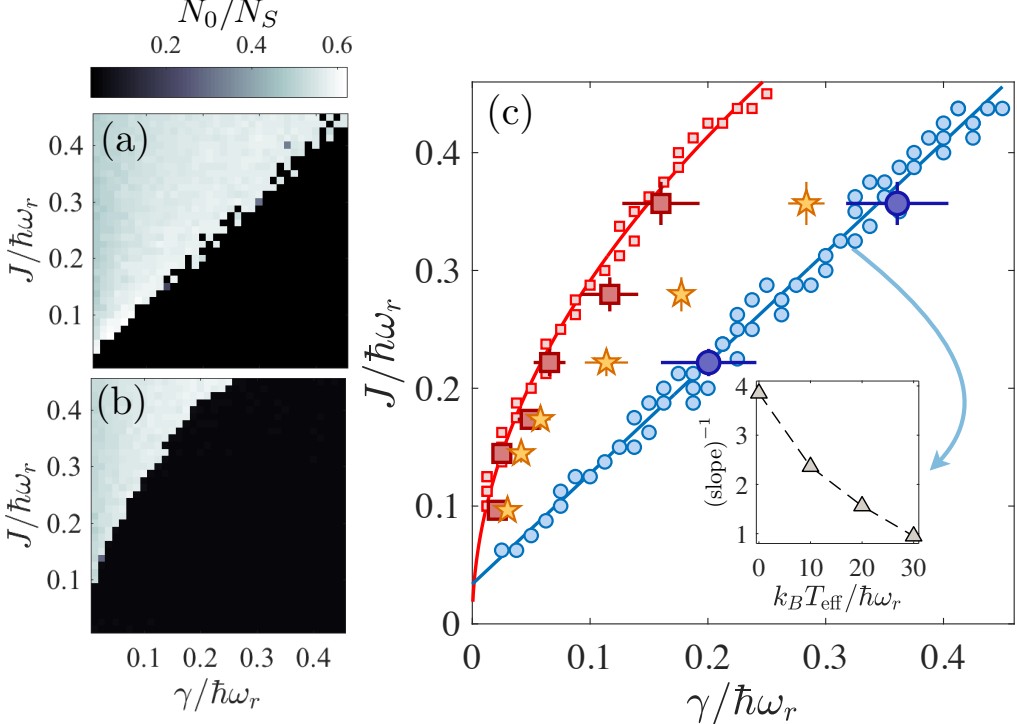

Figure 10: Steady-state condensate fractions $N_0/N_S$ for the multimode dynamical reservoir model in the (a) full and (b) empty branches in the $J$-$\gamma$ plane. This figure can be directly compared with Fig. 5 which did not include reservoir dynamics. (c) The phase diagram determined by numerically extracting phase boundaries from (a) and (b). The small, light colored markers show the boundaries extracted by detecting the boundary where the condensate fraction exceeded 0.4. The large, dark markers with error bars show the experimental data of Labouvie *et al.* [4]. The star markers show the observed values where critical slowing down occured in the experiment. The lines of best fit are $J = a\gamma + b$, $a = 0.94 \pm 0.02$, $b = 0.034 \pm 0.004$ for the superfluid branch, and $J = a\gamma^b$, $a = 0.94 \pm 0.02$, $b = 0.50 \pm 0.01$ for the lower branch. The inset in (c) shows the slope of the boundary for the superfluid branch obtained for a range of different effective temperatures.

## 6.2 Multimode dynamical reservoir model — Backaction and heating

The multimode dynamical reservoir model is able to *quantitatively* reproduce the nonequilibrium phase boundary scalings $\gamma_c \approx J$ and $\gamma_{LB} \propto \sqrt{J}$ with an appropriate choice of effective temperature for the reservoirs. However, this model shows only transient evidence of the nonequilibrium quasicondensate phase (unless $T_{\text{eff}} = 0$), and collapses to the normal phase at later times — We see no indication of critical slowing down within the bistable region, unlike in the experiment [see Fig. 10, orange stars]. We have however found quantitative agreement with the experimental observations for the other two phase boundaries by choosing only the chemical potential $\mu_R = 12\hbar\omega_r$ and the effective temperature $k_B T_{\text{eff}} = 30\hbar\omega_r$. However, an as-yet unexplained discrepancy is that our chemical potential is somewhat larger than what was in the experiment, for which we estimate the chemical potential to be approximately $\mu_R \sim 7\hbar\omega_r$ for the quoted atom number of $N \sim 700$ [4]. Our condensate number (which is cutoff independent and thus a meaningful measure) is $N \sim 990$.

It is important to note that all of the ingredients of finite temperature, independent reservoir dynamics, and back-action from the system site were all required to realise a reduction in

the critical dissipation rate to $\gamma_c \approx J$. Neglecting any one of these features results in a model with $\gamma_c \approx 4J$. The reduction of $\gamma_c$ is clearly due to a rather complicated combination of effects that all cooperatively reduce the effective value of $J$, including dephasing of $\psi_L$ and $\psi_R$, a reduced Frank-Condon factor $\eta$ [see Eq. (9)] from reduced coherence, and a reduction of the population in the driving sites ($N_R$ and $N_L$). Although the inset of Fig. 10(c) suggests that the effective temperature and dephasing are likely the dominant mechanisms that reduce $\gamma_c/J$, Fig 9(b) suggests that a counter-intuitive "parasitic condensation" phenomenon, whereby the coherence of the system is actually enhanced by the dissipation at the expense of the reservoir sites may also be a contributing factor.

### 6.3 Future work

Although the main focus of this work has been modelling a specific experiment with an atomic BECs [4], our predictions could potentially be explored in a coherently pumped exciton-polariton superfluid. Specifically, a confining potential could be created through mechanical stresses [28], and it could be driven by a non-uniform pump laser [36] that is blue-detuned from the bottom of the trap. The intermediate nonequilibrium quasicondensate regime presumably requires these features, as the modulational instabilities which give rise to pattern formation in the LLE seemingly do not occur in a uniform system described by Eq. (11) with $\beta > 0$ (see e.g. Ref. [38]). In the exciton-polariton context, stronger driving and dissipation could be considered than is possible in the ultracold atom setting, which may uncover additional phase regions similar to the tri- and pentastability regions observed in Ref. [37] for a simple dimer system. Given the success of the c-field approach to this damped-driven system, our c-field approach may yield further insight into the results of a related experiment [11], where the phenomenon of negative differential conductivity (which may have use in atomtronic applications [49–51]) was observed in the undamped refilling dynamics of an initially depleted site [9, 51]. By utilising a system with attractive interactions, or generalizing to multiple components, it may be possible to realize spontaneous pattern phenomena such as dissipative solitons [35], and Turing patterns [30, 52, 53], or Chimera states [40] in matter wave systems. It would also be of interest to extend the c-field model to include the full dynamics of all lattices sites and test the approximations made for the reservoirs in this paper, although this would be computationally challenging.

## 7 Conclusion

We have developed a c-field model of a multimode driven-dissipative Josephson array as experimentally realised by Labouvie *et al.* [4]. The basic features of bistability and hysteresis observed in experiment can be understood qualitatively from a single-mode mean-field approximation to the model where it is analytically tractable. Numerically simulating the full multimode problem, we have found that the nonequilibrium phase diagram qualitatively matches the results of Labouvie *et al.* [4]. Importantly, our model suggests that the observed critical slowing down in the lower branch is due to the formation of a nonequilibrium quasicondensate within a band of excited harmonic oscillator states, which are near-resonant with the driving frequency. While direct observation of the patterned nonequilibrium quasicondensate phase may be difficult experimentally, a clear signature appears in the atom number fluctuations, in which we observe a distinct peak at the transition. Meanwhile, a reduced robustness of the superfluid branch is obtained in our model by incorporating effects of reduced coherence and the backaction of the system on the reservoir sites.

Finally, we have identified that that the Lugiato-Lefever equation, widely used in nonlinear optics and exciton-polariton contexts, can also describe nonequilibrium dynamics of driven-

dissipative atomic superfluids. The degree of control available in ultracold atom systems allows for the tuning of parameters such as on-site energies, the sign of interactions, and could also introduce spin degrees of freedom. Further study of this system has the potential to realise as yet undiscovered nonequilibrium states of quantum matter.

# Acknowledgments

We are particularly grateful to Herwig Ott for discussions and feedback on drafts of this manuscript. We also thank Ewan Wright, Brian Anderson, Ashton Bradley, Samuel Begg, and Lewis Williamson for further discussions and input.

**Funding information**  This research was supported by the Australian Research Council Centre of Excellence in Future Low-Energy Electronics Technologies (project number CE170100039) and funded by the Australian Government. M.T.R is supported by an Australian Research Council Discovery Early Career Researcher Award (DECRA), Project No. DE220101548.

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
