# Peer review of "Bistability and nonequilibrium condensation in a driven-dissipative Josephson array: a c-field model"

_SciPost Physics, doi:SciPost Phys. 15, 068 (2023)_

## Round 2 · Referee Report · Anonymous (Referee 1) · 2022-12-8

Strengths

  • Very clearly written
  • Thorough analytical and numerical analysis,
  • Discussion also of the limitations of the approaches used
  • The models provide possible explanations of the experimentally observed effects

Weaknesses

  • Fine tuning and ad hoc assumptions about the reservoirs are needed to obtain better agreement with experiment

Report

In their paper “Bistability and nonequilibrium condensation in a driven-dissipative Josephson array: a c-field model” Reeves and Davis present a theoretical analysis of an intriguing recent experimental study by the group of Herwig Ott. In this experiment an array of Joesphson junctions is realized using weakly interacting ultracold bosonic atoms in a one-dimensional optical lattice of two-dimensional pancake-shaped traps (sites), each of which hosts a number of relevant transverse states. The system is additionally subjected to controlled local particle loss at the central lattice site. For a wide interval of dissipation strengths, a bistable region is found, where, depending on whether the central dissipative site was initially empty or filled, it remains weakly populated or entirely filled, respectively. While the latter case suggests that a supercurrent into the lossy site is established, the former is interpreted as a regime of incoherent transport. Moreover, it is found that the time the system needs to reach its (quasi)steady state in the normal (non superconducting) branch shows a pronounced maximum as a function of the dissipation strength. This observation was interpreted as possible critical slowing down indicating a phase transition inside the bistable region.

In their paper Reeves and Davis present and solve three different models of increasing complexity to describe the experiment. The first and most simple model, which is introduced to provide a first intuitive picture and reference point, treats the lossy lattice site in a single-mode approximation. It can be solved analytically and shows indeed bistability. Second, a multi-mode model of the lossy site is introduced, using the c-field approach. Here (roughly) only those modes are kept, which, thanks to an occupation larger compared to one, allow for a classical description. This leads to a stochastic Gross-Pitaevskii equation, which is projected onto the (low-energy) subspace spanned by the relevant modes. Here the remaining lattice is captured by a driving term. This approach equally gives rise to bistability. Moreover, it provides also a possible explanation of the transition (or crossover) and critical slowing down happening in the bistable region. Namely, it is associated with the formation of a large occupation of several excited modes which are in resonance with the reservoir (which the authors call quasicondensate). While offering an explanation for various experimentally observed features, the single-site c-field model fails to provide quantitative agreement. Therefore, it is extended by making the reservoir dynamical. Both the left and the right reservoir are treated using a phenomenological multi-mode model coupled to an effective thermal reservoir. By tuning the effective temperature of the bath, quantitative agreement with the experiment can be established for this third model. However, this model does not give rise anymore of the tradition (“quasicondensation”) within the bistable region.

The paper provides a thorough analysis of the non-equilibrium behavior of a dissipative quantum system. It is very well written. The authors provide the right amount of details (not too few not too many) and the way the paper is structured from simple to more elaborate models works well. Also the limitations of the methods used are discussed. The discussion of the models and their properties (obtained analytically and using the c-field approach) is well organized and clear. Remaining discrepancies between theory and experiment as well as ad hoc assumptions regarding the design of the dynamical reservoirs and the need to fine tune the effective temperature to achieve better agreement suggest, however, that the models do not yet fully explain the physics of the experimental system. But his is fine. I am convinced that the paper provides interesting insights into the physics of the system and possible mechanisms explaining its features. Before recommending publication, I would, however, kindly ask the authors to address the following (mostly minor) points.

Requested changes

1) Is it obvious to base the truncation on the enemies of the single-particle orbitals? For Bose condensates in a trap it is well known that the Thomas-Fermi radius can differ substantially from the harmonic oscillator ground state. Does the Thomas Fermi-Radius provide a rough estimate for the number of states needed to be taken into account?

2) I do not link the use of the term “quasicondensate” very much. This word has been introduced for specific superfluid equilibrium states, featuring quasi-long range order (i.e. algebraic decay of the single-particle coherences). Maybe there is a better way of calling this state.

3) I find the non-equilibrium state which is called quasicondensate very interesting. I think it would be desirable, if the authors could provide more details about it and how the transition/crossover to the normal phase occurs.

4) Please specify the gamma values used in the right panels of Fig. 3. (Here also a series of such plots showing the transition from normal via “quasicondensate” to condensate at more intermediate points would be very interesting, see 3).

5) Maybe g^(2) should be called two-particle correlations (rather than coherences).

6) In Fig. 5.(c) the lower phase boundary the label SF/ NS seems to be wrong.

7) I think the choice of the effective temperature should be explained in more detail. Is there any rational behind choosing a particular values, besides tuning the phase boundary to the experimental value?

8) Maybe, I overlooked it, but I did not find information about how the yellow star symbol in Fig. 10 were obtained. Probably they give information about the metastable “quasicondensation”. I think it would be very nice to see some data indicating this transient phenomenon.

---

## Round 2 · Referee Report · Anonymous (Referee 2) · 2023-2-12

Strengths

1- Very comprehensive 2- High quality and clear figures (although see comments on figure 2 below) 3- A decent effort put in to make the paper both precise and accessible.

Weaknesses

1- Some loose terminology (see below for comments)

Report

This is a comprehensive piece of work, carefully put together by experts in the field. There is a good balance of specificity (with respect to a particular experiment) and general theoretical development. It is quite long, containing quite a lot of material, and such papers can be challenging to write. I have a number of suggestions/requests for improvements which are essentially all in the nature of improving clarity and tightening up some terminology. However it is a good paper and should be published.

Requested changes

1- Page 1, column 2, paragraph 2: "numerical explore its features" should be "numerically explore its features"

2- Page 2, column 2, equation 1: I think it would be preferable to at least refer to/explain dW as a Wiener noise term a bit sooner after it is introduced.

3- Page 3, column 1, paragraph 2: It is a "Wiener noise term", not "Weiner".

4- Page 4, column 1, paragraph 3: "one or three solutions" could use a bit more precision. A cubic equation always has 3 solutions, however there is a physically motivated criterion here that only non-negative real solutions are meaningful, and it turns out that there is always one and there may be three. I'll also note that I'm a bit surprised that the algebraic solutions aren't just presented, at least in an appendix; they should be fairly straightforward to determine (there's a fairly decent description of the process even on Wikipedia!).

5- Page 4, column 1, equation 19: I think it would be clearer if it was stated that Eq. (18) was differentiated with respect to n_S, and then d J^2/dn_S were set = 0.

6- Page 4, column 2, figure 2: this should be considered optional, however I did note that the various annotations on these plots seemed to be a somewhat inconsistent mixture of times new roman and computer modern (i.e., default LaTeX) fonts. On (d) the ticking on the y axis does not appear even to be the same size as that own the x axis.

7- Page 5, column 2, section IV, paragraph 2: Another optional suggestion. It might be advisable to refer to this method of determining the condensate as an Onsager-Penrose approach; it's not strictly identical to e.g. a symmetry-braking approach of averaging over a field operator.

8- Page 9, column 1, paragraph 1: I am very unsure over the nomenclature "instantaneous chemical potential"! Chemical potential, like temperature, is a strictly equilibrium concept. I think a better way of thinking about what it is that there is a functional of psi which, when psi is a stationary state, gives a value that is a chemical potential. For sufficiently slow timescales, it may be that the concept that a local (in time) "instantaneous" chemical potential is a useful concept, but it's not clear that that is the case here. I would also note that the subscript j appears on the right hand side of the equation, but not the left. Is this an oversight, or is there a summation convention?

9- Page 9, column 2, paragraph 2: Just a comment here regarding "A possible explanation for this outcome is that the coherent condensate atoms tunnel more easily into the system"; this may or may not be a useful analogy, but I cannot help thinking of the superfluid (but not the normal fluid) flowing through a capillary in HeII.

10- page 10, column 1, paragraph 1: I would suggest "stationary states of the GPE" (or something of the sort) as being more appropriate than "eigenmodes".

---

## Round 3 · Referee Report · Anonymous (Referee 1) · 2023-5-5

Report

As I pointed out in my previous report, the manuscript provides a very well written and carefully conducted theoretical and numerical study of an intriguing experiment. While it still not fully explains every aspect of the experiment, it provides new insights into possible mechanisms explaining the experimental data. Since, moreover, the authors addressed the (minor) points raised in my previous report in a satisfactory fashion, I am happy to recommend the publication of the revised manuscript in SciPost Physics in its present form.

---

## Round 3 · Referee Report · Anonymous (Referee 2) · 2023-5-9

Report

The authors have responded very creditably to the comments of both referees, and I am happy to approve the manuscript as it stands.

---

## Round 3 · Author Response

Dear Editor,

We are glad to see that the reviewers feel the manuscript struck the right balance between modelling the specific experiment and a general theoretical development, as was our aim for this work. We thank the reviewers for their comprehensive and broadly positive assessment of our work, and their helpful comments/suggestions.

We have made changes to address all of the minor points raised by the reviewers, as detailed below. We believe the manuscript is now suitable for publication.

---

## Round 3 · List of Changes

Changes in Response to Reviewer 1

  1. The number of single-particle states that need to be considered is determined by the chemical potential. So, yes --- equivalently, one could use the Thomas Fermi radius, since they are related via the relation R = \ sqrt(2 mu/ m omega^2). As detailed in the manuscript, for the problem at hand, a cutoff in the range 2-3 mu was found to be appropriate. To address this point, we have expanded the explanation of the projector after Eq. (4). We have also included additional references, which explain the rationale of choosing the single particle basis states to define the projector / cutoff energy.

  2. We have relabelled the "quasi condensate (QC)" state as a “nonequilibrium quasi-condensate (NQC)” throughout the text and all the figures, to help distinguish this nonequilibrium steady state from the traditional (equilibrium) “quasi-condensate”.

  3. Expanded the text on pg. 7 explaining how the NQC state emerges from the normal state.

  4. Added the parameter values to the caption of Fig. 3. The additional text provided in response to point 3 also gives additional information about the emergence of the non equilibrium quasi-condensate from the normal state.

  5. We have termed g_2 as the "second order correlation function" throughout. However, we still refer to density and phase "coherences" when referring to g_1 and g_2 respectively, as this is physically what these correlation functions measure.

  6. No change (the figure labels are correct). In the region bounded by the blue circles and the orange stars, the system is in the SF state when initialized on the full branch (a), and is in the normal state when initialized in the empty branch (b). This can be seen by comparing the values of the condensate fraction presented in Figs 5a and 5b within the region bounded by those two boundaries.

  7. For our purposes the effective temperature is simply a phenomenological fitting parameter; it incorporates additional sources of noise (thermal, etc.), but there is no rationale for choosing a particular value besides tuning the phase boundaries. We have added additional text after Eq. (30) explaining this point further.

  8. Added the following to the caption of Fig. 10: “The star markers show the observed values where critical slowing down occured in the experiment.” We also added the following text on pg. 10: “For completeness, in Fig. 10 we also show the points where critical slowing down was observed in the experiment [orange stars], although no signature of this boundary appears in our dynamical reservoir model.” As we suggest at the end of the discussion, we suspect that simulating the dynamics of the full chain may be needed to capture this boundary correctly.

Changes in Response to Reviewer 2:

  1. Corrected typographical error

  2. Added the following text after Eq. (1): “L is the Gross-Pitaevskii operator, γ is the particle loss rate, F is the driving associated with the reservoir sites, and dW is a Wiener noise term associated with the losses.” We then go into the specifics of each term.

  3. Corrected typographical error

  4. We have changed the text to read “Eq. (18) Permits either one or three solutions (real and positive) …”. We also agree that some of the explanation in this section was probably not required; in the interest of brevity, we have therefore also shortened the explanation throughout this section by removing several remarks that were not essential.

  5. We have changed the text to read “differentiating Eq. (18) with respect to n_S yields …”

  6. We have ensured the x and y ticks in Fig. 2(d) are the same size and that the figures have consistent fonts throughout the manuscript.

  7. We have added the following text in Sec. IV, pg 5: “To verify that our results did not depend on the choice of cutoff, we calculated the condensate number $N_0$, as determined from the Onsager-Penrose criterion [31], which specifies the condensate number as the largest eigenvalue of the one-body density matrix”.

  8. We agree that the term “instantaneous chemical potential” is a loose concept and this interpretation is only valid under certain conditions and assumptions. We have rephrased the text after Eq. (32) as follows: “The quantity $\bar \mu$ may be loosely interpreted as an “instantaneous chemical potential”, although strictly the chemical potential is only defined in equilibrium.” Also, there is no summation convention; for clarity we have therefore added the subscript $j$ to the quantity $\bar \mu$ in Eq. (32).

  9. We think this is an interesting observation and have added the following text in the relevant paragraph: “Similar behaviour is well known in the study of superfluid helium through capillaries, wherein the superfluid can easily flow but the normal fluid cannot.”

  10. We have changed the text to the following: “the system can still be reduced to a single mode in terms of the stationary states of the nonlinear GPE operator $\mathcal{L}$

---

## Editorial Decision

published